# Assessment of Survival Kinetics for Emergent Highly Pathogenic Clade 2.3.4.4 H5Nx Avian Influenza Viruses

**DOI:** 10.3390/v16060889

**Published:** 2024-05-31

**Authors:** Caroline J. Warren, Sharon M. Brookes, Mark E. Arnold, Richard M. Irvine, Rowena D. E. Hansen, Ian H. Brown, Ashley C. Banyard, Marek J. Slomka

**Affiliations:** 1Virology Department, Animal and Plant Health Agency (APHA-Weybridge), Woodham Lane, Addlestone, Surrey KT15 3NB, UK; sharon.brookes@apha.gov.uk (S.M.B.); richard.irvine002@gov.wales (R.M.I.); rowena.hansen@apha.gov.uk (R.D.E.H.); ian.brown@apha.gov.uk (I.H.B.); ashley.banyard@apha.gov.uk (A.C.B.); 2Department of Epidemiological Sciences, Animal and Plant Health Agency, Sutton Bonington, Loughborough LE12 5RB, UK; mark.arnold@apha.gov.uk; 3Office of the Chief Veterinary Officer (OCVO), Welsh Government, Cathays Park, Cardiff CF10 3NQ, UK; 4Veterinary Advice Services, Animal and Plant Health Agency (APHA-Weybridge), Woodham Lane, Addlestone, Surrey KT15 3NB, UK; 5WOAH/FAO International Reference Laboratory for Avian Influenza, Animal and Plant Health Agency, (APHA-Weybridge), Woodham Lane, Addlestone, Surrey KT15 3NB, UK

**Keywords:** avian influenza virus, clade 2.3.4.4, avian, survival, kinetics

## Abstract

High pathogenicity avian influenza viruses (HPAIVs) cause high morbidity and mortality in poultry species. HPAIV prevalence means high numbers of infected wild birds could lead to spill over events for farmed poultry. How these pathogens survive in the environment is important for disease maintenance and potential dissemination. We evaluated the temperature-associated survival kinetics for five clade 2.3.4.4 H5Nx HPAIVs (UK field strains between 2014 and 2021) incubated at up to three temperatures for up to ten weeks. The selected temperatures represented northern European winter (4 °C) and summer (20 °C); and a southern European summer temperature (30 °C). For each clade 2.3.4.4 HPAIV, the time in days to reduce the viral infectivity by 90% at temperature T was established (D_T_), showing that a lower incubation temperature prolonged virus survival (stability), where D_T_ ranged from days to weeks. The fastest loss of viral infectivity was observed at 30 °C. Extrapolation of the graphical D_T_ plots to the x-axis intercept provided the corresponding time to extinction for viral decay. Statistical tests of the difference between the D_T_ values and extinction times of each clade 2.3.4.4 strain at each temperature indicated that the majority displayed different survival kinetics from the other strains at 4 °C and 20 °C.

## 1. Introduction

Avian influenza viruses (AIVs) are classified into subtypes according to the external viral glycoproteins, namely the haemagglutinin (HA; H1-H16) and neuraminidase (NA; N1-N9) [1]. The H5 and H7 AIV subtypes are notifiable avian disease agents [2,3], and can mutate from the low pathogenicity (LP)AIV to the corresponding high pathogenicity (HP)AIV, with the latter causing high morbidity and mortality in poultry. Wild birds are the natural reservoir for AIVs [4], with migration enabling the maintenance and dissemination of a diverse array of AIV subtypes, including clade 2.3.4.4 HPAIVs of the “goose/Guangdong” (GsGd) H5 HPAIV lineage that have been decimating avian populations globally in recent years. The GsGd lineage first emerged in 1996 with subsequent evolution through diverse clades [5,6,7,8], followed by intercontinental spread via wild birds [9,10]. Clade 2.3.4.4 epizootics have impacted poultry production systems, local economies, and international trade [11].

Since 2014, multiple H5Nx HPAIV clade 2.3.4.4 incursions have occurred in Europe, with outbreaks increasing in frequency since 2020 [12,13,14]. The H5N8 subtype dominated the winter 2020–2021 epizootic and persisted into summer 2021, before significant re-emergence during autumn 2021 as the unique H5N1 subtype, albeit as multiple genotypes, for the ensuing winter [14]. This clade 2.3.4.4b H5N1 HPAIV persisted through summer 2022 [15], followed by numerous poultry outbreaks and wild bird cases during winter 2022–2023 [16]. Since 2020, these European clade 2.3.4.4b H5Nx epizootics have demonstrated that viruses circulating in wild birds represent the greatest threat to the poultry sector [13,14].

AIV survival in different environments is a critical component of understanding incursion risk to different poultry sectors [17]. The survival of infectious AIVs has been demonstrated in water, soil, and faeces [18,19,20,21], and in different egg products [22]. Multiple factors contribute to AIV survival metrics [23], hence comparisons in distilled water and saline [24,25], plus the effects of pH and temperature have featured in assessments of viral decay [26,27]. In this study, we investigated the survival kinetics for five clade 2.3.4.4 H5Nx HPAIVs linked to UK incursions which were incubated at up to three temperatures.

## 2. Materials and Methods

### 2.1. Viruses and Safety Statement

Five H5Nx clade 2.3.4.4 HPAIV isolates derived from outbreak events in the United Kingdom (UK) were investigated in this study (Table 1) as representatives of viruses which caused disease events during five UK/European clade 2.3.4.4 epizootic seasons between 2014 and 2021. Viruses were propagated in 9- to 11-day-old specified pathogen-free embryonated fowls’ eggs (SPF EFEs) [28]. The H5Nx clade 2.3.4.4 HPAIVs are categorised in the UK by the Specified Animal Pathogens Order at level 4 and the Advisory Committee on Dangerous Pathogens at level 3, hence all laboratory work was carried out in licensed biosafety level 3 laboratories [29].

### 2.2. Cell Lines

Madin-Darby canine kidney (MDCK) cells, supplied originally from the European Collection of Cell Cultures (ECACC, Genova, Italy), were propagated at 37 °C using Eagle’s minimal essential medium (EMEM) with 10% (*v*/*v*) foetal bovine serum (FBS), which contained antibiotics (penicillin 100 IU/mL and streptomycin 100µg/mL; Life Technologies Ltd., Carlsbad, CA, USA), in a 5% (*v*/*v*) CO_2_ atmosphere.

### 2.3. Preparation of Replicate Samples for Infectivity Assessment at Different Temperatures

Each virus stock was diluted in serum-free medium to prepare a working concentration at a 50% tissue culture infectious dose (TCID_50_) of 1 × 10^6^ TCID_50_/mL (Table 1). At time zero, three replicate samples (1.2 mL) were prepared, using sterile safe lock Eppendorf tubes (Anachem, Clonakilty, Ireland), for each investigated timepoint at a given temperature (detailed below). An additional eight to 10 aliquots of the working concentration of each virus were prepared and frozen promptly, to serve as time zero positive controls representative of no incubation treatment.

All five H5Nx isolates were incubated at 4 °C (±2 °C), and 20 °C (±2 °C), with three incubated at a higher temperature 30 °C (±2 °C), using an LR902 refrigerator (LEC Refrigeration plc., Prescot, UK) or a 305NP heat-cooled incubator (LMS Ltd., Sevenoaks, UK). Both instruments featured independent temperature monitoring (Traceable Wireless Sensor, Tutela Medical.com (https://www.tutelamedical.com, accessed on 23 May 2024). At each timepoint and incubation temperature, three biological replicates were removed and promptly transferred to −85 °C storage. The timepoint collections were as follows: at 4 °C on days 0, 1, 2, 3, 4, 7, 9, 11, and 14, thereafter weekly to ten weeks; at 20 °C on days 0, 1, 2, 3, 4, 7, 9, 11, and 14, and weekly to seven weeks; and at 30 °C on days 0, 1, 2, 3, 4, 5, 6, and 7, and twice each week for up to five weeks.

### 2.4. Virus Quantification after Incubation at Different Temperatures

The infectivity of each timepoint replicate was quantified as a 50% TCID_50_ value of 96-well plates (Nunc, ThermoFisher, Waltham, MA, USA) which contained 80% confluent MDCK monolayers. Virus survival was determined as the time (days) for the viral infectivity to be reduced by 90% at the incubation temperature T (D_T_), or a one log_10_ reduction. Before titration, the 96-well MDCK plates were washed in 100 µL of serum-free EMEM and replaced with 100 µL of fresh media. Once each stored biological replicate had thawed, 100 µL of media was removed from all wells in the first column of a 96-well MDCK plate, with all remaining wells containing 100 µL of serum-free EMEM. Each biological replicate was divided between wells in column 1 and applied as eight 146 µL volumes (technical replicates). Forty-six µL volumes were removed from the first column and diluted into the second column with titration repeated sequentially from column 2 into column 3 and continued across to column 11 to produce a 0.5 log_10_ dilution series. As an internal negative control, no virus was titrated into column 12. For the positive virus control (no incubation treatment), one time zero frozen aliquot was thawed, and applied for quantification in a separate 96-well plate of MDCK cells. After incubation at 37 °C (±2 °C) for 60 min, microtiter plates were washed using 100 µL of serum-free EMEM, before being overlaid with 100 µL of fresh serum-free EMEM. Each plate was incubated at 37 °C (±2 °C) and 5% CO_2_ for up to four days, for the cytopathic effect to develop.

Following incubation, the medium was removed, and all wells were washed in 100 µL of serum-free EMEM and blotted before 50 µL of crystal violet solution (1% (*w*/*v*) in water: ethanol (1:5.3); Sigma-Aldrich, St. Louis, MI, USA) was added to each well. After 30 min incubation at room temperature, the crystal violet solution was discarded, the wells were washed using 100 µL of 0.1 M phosphate-buffered saline (pH 7.2), and the plates were blotted dry. All titrations of biological replicate samples were performed in triplicate and a TCID_50_/mL was determined for each from the mean of the eight technical replicates assessed [30]. The limit of detection for virus infectivity was a virus titre of 1.625 TCID_50_/mL, below which a zero titre was recorded.

### 2.5. Statistical Analyses

Univariate linear regression analyses [24], were undertaken for each biological sample at each temperature treatment. Values for the viral log_10_ TCID_50_/mL were plotted against time in days to determine viral decay kinetics at a chosen temperature. The slope of the corresponding line of best fit was used to calculate D_T_ for a 90% reduction in viable viral population [31]. Since three biological replicates per timepoint and temperature profile were prepared from the same working stock, each 50% TCID_50_ calculation represented an independent test. To compare the best graphical fit for virus survival against time, the extra-sum-of-squares F test was used to determine how well these observed values fitted those expected from a regression analysis. Deviation of the slope from zero was considered statistically significant at the 5% level when *p* < 0.05. The R^2^ values reflected the fit of these data to the regression line. Extrapolation of the straight line to the x-axis intercept (y = 1 log_10_ TCID_50_/mL) gave a corresponding time to extrapolated extinction (days) for a given H5Nx HPAIV at the stated incubation temperature. The statistical significance of differences in the D_T_ and extinction times between isolates was carried out using the analysis of covariance tool in Matlab v2019b (MathWorks, Natick, MA, USA), adjusted for multiple comparisons using a Tukey test. All other statistical analysis were performed using GraphPad Prism, version 8.4.2 (GraphPad Software, La Jolla, CA, USA).

## 3. Results and Discussion

The ability of AIV infectivity to survive in the environment is a major contributing factor to viral persistence and spread [17], and merited investigation because of the extensive nature of the most recent H5N1-2021 clade 2.3.4.4b epizootic [16]. Experimental evidence has underlined the essential waterfowl-adapted tropism of H5Nx clade 2.3.4.4 HPAIVs, where the efficient acquisition and onward transmission of infection correlated with waterfowl interactions, within a strongly virus-contaminated environment [32,33,34,35,36]. This sustained viral maintenance within wild anseriformes produces the infection pressure for associated poultry outbreaks [7], with viral environmental contamination also recorded at outbreak premises [37,38]. We compared the temperature-associated virus stability (infectivity) of five UK-origin H5Nx clade 2.3.4.4 strains (Table 1) at three European outdoor temperatures, where 4 °C mirrored winter, while 20 °C and 30 °C corresponded to summer temperatures in northern and southern Europe, respectively. Viral stability was determined from the D_T_ values at a given temperature [24], following incubation for up to 10 weeks. Extrapolation of the predicted straight line provided an intercept to also show the time to extinction of viral infectivity (Figure 1 and Figure 2, Table 2). Each clade 2.3.4.4 isolate showed greatest stability at 4 °C, whereas increased incubation temperatures, namely 20 °C and 30 °C, resulted in a faster reduction in viral infectivity (Figure 1 and Figure 2, Table 2), which was consistent with an inverse relationship for AIV survival kinetics and temperature [24,39].

For the five viruses assessed at 4 °C and 20 °C, pairwise comparisons of their D_T_ values and extrapolated extinction times revealed that, in the majority of cases, the virus survival (stability) of each virus was significantly different from the others (*p* < 0.05; Table 3 and Table 4). The only exceptions among the 4 °C D_T_ comparisons were those for H5N8-2014 versus H5N1-2021 (*p* = 0.33), and H5N8-2016 versus H5N6-2017 (*p* = 0.06), with comparison of H5N1-2021 versus H5N6-2017 being non-significant for both their D_T_ values (*p* = 0.22) and extinction times (*p* = 0.58; Table 3). Exceptions among the 20 °C comparisons were noted for the D_T_ values for H5N1-2021 versus H5N8-2014 (*p* = 0.55), for the extinction times for H5N1-2021 versus H5N6-2017 (*p* = 0.13), with comparison of H5N6-2017 versus H5N8-2014 being non-significant for both their D_T_ values (*p* = 0.35) and extinction times (*p* = 0.97; Table 4). However, comparisons of both parameters obtained at 30 °C showed that only H5N8-2016 had a significantly longer extinction time than H5N8-2020 and H5N1-2021 (*p* < 0.001), with no significant difference between the extinction times of H5N8-2020 and H5N1-2021 (*p* = 0.33). There were no significant differences between the D_T_ values among any of the three isolates at 30 °C (*p* > 0.28).

Overall, for a given H5Nx clade 2.3.4.4 isolate, these data showed the reduction in survival time at 20 °C was at least 2.5-fold faster than the D_T_ values observed at the low incubation temperature (4 °C). Wild waterfowl cases, during the clade 2.3.4.4 epizootic waves, peaked in the European winter months when these heightened infection pressures would result in accompanying incursions in farmed poultry [12,16]. The two least stable viruses, at least when ranked by their extinction times, were H5N8-2014 and H5N6-2017 (Table 2), and their corresponding significance tests at 20 °C (Table 4) showed this virus pair was significantly different from the other viruses. Interestingly, the European winter incursions of H5N8-2014 and H5N6-2017 were the smallest of the five H5Nx clade 2.3.4.4 epizootics, which in the case of H5N6-2017 were essentially limited to wild birds with no accompanying commercial poultry outbreaks during winter 2017–2018 [12,40]. The relative instability of H5N8-2014 and H5N6-2017, reflected in low D_T_ and extinction values, may have contributed to the limited scale of these incursions.

In field settings, there is the confounder of matrix effects upon virus deposited in the environment, which may unpredictably affect the complexity of virus survival. Therefore, it is important to apply uniform and consistent assessment of temperature-associated survival of any viral pathogen which is excreted into the environment, hence all five H5Nx HPAIVs were each isolated and propagated by using SPF EFEs. Progeny AIVs are produced during the orthomyxovirus replication cycle [8,41], during which they acquire an external lipid envelope of plasma membrane origin from the host [42]. Therefore, lipid envelope consistency was ensured by focusing the temperature stability investigations on viruses propagated in EFEs. This approach ensured the variation in infectivity survival was determined by mainly differences among the viral envelope proteins, namely the HA and NA [23,43]. However, potential contributions to viral stability by other internal structural proteins could not be entirely excluded [23,44], and may also extend to the stability of the encapsulated viral polymerase enzymes, so these could have contributed to the stability differences observed in our study for the three H5N8 clade 2.3.4.4 subtypes (Table 2).

Temperature-associated viral stability reflected in environmental survival is but one factor which may influence the scale and extent of epizootics. The recent H5N1 clade 2.3.4.4b HPAIV epizootic has included an expansion of the avian host range to include seabirds with many cases reported on a global scale, thereby enhancing viral spread and associated infection pressure [10,15,45]. However, the dynamics of spread within waterfowl may be modulated by host responses, including innate and/or H5-specific humoral immunity acquired during previous AIV incursions [46,47]. Other species–specific differences have been noted, where environmental contamination may not have such a prominent role in H5Nx clade 2.3.4.4 HPAIV spread among chickens [48,49].

## 4. Conclusions

These quantified virus survival data provide evidence to refine disease prevention strategies for poultry units and inform future veterinary risk assessments for outbreak management, particularly in view of the continuing H5N1 clade 2.3.4.4 HPAIV global epizootic [50,51]. Importantly, these virus survival outcomes contribute to the understanding of AIV persistence in the environment, thereby informing protection of poultry health and commercial production systems. These data highlight HPAIV persistence at low temperatures, so presenting a greater infection risk to avian species during the cooler months in temperate latitudes, especially in the case of clade 2.3.4.4 H5Nx HPAIVs to waterfowl with subsequent incursion risks for farmed poultry. Our statistical data, using pairwise comparisons of virus D_T_ values and extrapolated extinction times at 4 °C and 20 °C, (*p* < 0.05; Table 3 and Table 4), revealed that, in many instances, the virus survival (stability) of each isolate was significantly different from the others. The relevance of these experimental findings has been underlined by the detection of H5N1 HPAIV in the immediate farm environment during UK clade 2.3.4.4 outbreaks in 2023 [38], affirming observations during earlier H5N1 GsGd HPAIV outbreaks [37]. These experimental and field-based studies are now identifying the consequences of environmental contamination, which have arisen from AIV incursions (either wild birds or poultry) and may influence onward spread. The temperature stability of isolates can also inform the likelihood of continuing infectivity, particularly during the vulnerable period prior to statutory cleansing and on-farm disinfection interventions [52,53]. Interestingly, heat treatment may provide an alternative to chemical disinfection, thereby giving additional importance to the outcomes of viral temperature stability investigations [54,55].

## Figures and Tables

**Figure 1 viruses-16-00889-f001:**
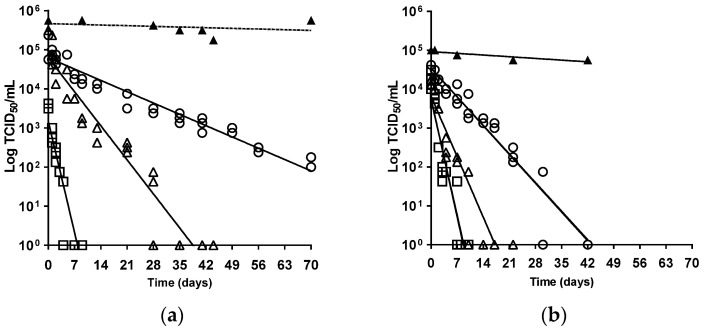
Linear regression for virus kinetics profiles for (**a**) H5N8-2020; (**b**) H5N1-2021 survival on MDCK cell monolayers in serum-free EMEM with penicillin and streptomycin. Logarithm TCID_50_/mL was plotted versus the time in days at 4 °C (±2 °C), 20 °C (±2 °C), and 30 °C (±2 °C). Symbols: ⚪ at 4 °C, ∆ at 20 °C, □ at 30 °C, ▲ positive controls (no incubation treatment) assessed per virus per batch of MDCK cell monolayers. For all isolates evaluated, mean R^2^ 0.91, *p* < 0.0001.

**Figure 2 viruses-16-00889-f002:**
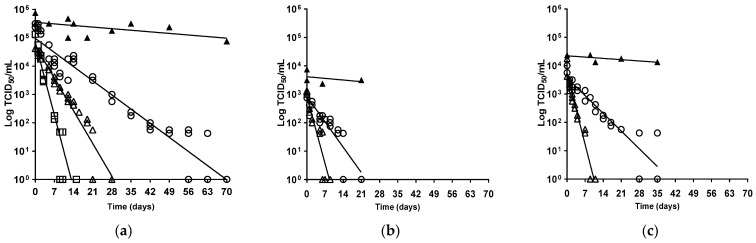
Linear regression for virus kinetics profiles for (**a**) H5N8-2016; (**b**) H5N8-2014; (**c**) H5N6-2017 survival, on MDCK cell monolayers in serum-free EMEM with penicillin and streptomycin. Logarithm TCID_50_/mL was plotted versus the time in days at 4 °C (±2 °C), 20 °C (±2 °C), and 30 °C (±2 °C). Symbols: ⚪ at 4 °C, ∆ at 20 °C, □ at 30 °C, ▲ positive controls (no incubation treatment) assessed per virus per batch of MDCK cell monolayers. For all isolates evaluated, mean R^2^ 0.91, *p* < 0.0001.

**Table 1 viruses-16-00889-t001:** Summary of the five UK-origin H5Nx clade 2.3.4.4 HPAIVs assessed for temperature stability.

Isolate	Subtype/Clade	Descriptor in Text	Collection Date ^1^
A/duck/England/1279/2014	H5N8 2.3.4.4a	H5N8-2014	16 November 2014
A/wigeon/Wales/52833/2016	H5N8 2.3.4.4b	H5N8-2016	14 December 2016
A/swan/England/001986/2017	H5N6 2.3.4.4b	H5N6-2017	31 December 2017
A/chicken/England/030786/2020	H5N8 2.3.4.4b	H5N8-2020	1 November 2020
A/chicken/Wales/053969/2021	H5N1 2.3.4.4b	H5N1-2021	31 October 2021

^1^ Collection dates of the five UK H5Nx clade 2.3.4.4 HPAIV isolates were sourced from the GISAID EpiFlu™ database (https://www.epicov.org/epi3/frontend accessed on 23 May 2024).

**Table 2 viruses-16-00889-t002:** Survival assessment of five H5Nx HPAIV isolates at up to three different temperatures. Results are expressed as D_T_ time and the extrapolated time to extinction (both as days) and ranked according to the latter for a given temperature. Survival kinetics experiments at 4 °C (±2 °C) lasted to ten weeks, at 20 °C (±2 °C) to seven weeks (combined mean R^2^ 0.92, *p* < 0.0001), and at 30 °C (±2 °C) to five weeks (mean R^2^ 0.90, *p* < 0.0001).

Temperature	4 °C		20 °C			30 °C	
Strain	D_T_	Extinction ^1^	D_T_	Extinction ^1^	Strain	D_T_ (days)	Extinction ^1^
H5N8-2020	24.2	116.1	8.0	38.5	H5N8-2016	2.7	13.2
H5N8-2016	14.0	69.8	6.1	28.8	H5N1-2021	2.3	8.6
H5N1-2021	9.5	43.2	4.5	17.2	H5N8-2020	2.5	7.8
H5N6-2017	11.2	40.0	2.8	10.4			
H5N8-2014	7.8	22.9	3.0	8.7			

^1^ Extinction (days) is the x axis intercept (y = 1 log_10_ TCID_50_/mL).

**Table 3 viruses-16-00889-t003:** Resulting *p*-values from significance tests of the difference between the D_T_ values and extinction times at 4 °C for five different H5Nx HPAIV isolates.

HPAIV	Result of Multiple Comparisons of D_T_ Value/Extinction Time at 4 °C				
HPAIV Isolate	H5N8-2014	H5N8-2016	H5N6-2017	H5N8-2020	H5N1-2021
H5N8-2014	-	<0.05/<0.001	<0.001	<0.001	0.33/<0.001
H5N8-2016	-	-	0.06/<0.001	<0.001	<0.001
H5N6-2017	-	-	-	<0.001	0.22/0.58
H5N8-2020	-	-	-	-	<0.001
H5N1-2021	-	-	-	-	-

A single *p*-value cell means that the same *p*-value applies to both the D_T_ value and extinction time. - indicates that either (i) table diagonals: result not applicable as it is for the virus strain versus itself, or (ii) lower half of table: result available in the top half of the table.

**Table 4 viruses-16-00889-t004:** Resulting *p*-values from significance tests of the difference between the D_T_ values and extinction times at 20 °C for five different H5Nx HPAIV isolates.

HPAIV	Result of Multiple Comparisons of DT Value/Extinction Time at 20 °C				
HPAIV Isolate	H5N8-2014	H5N8-2016	H5N6-2017	H5N8-2020	H5N1-2021
H5N8-2014	-	<0.01/<0.001	0.35/0.97	<0.001	0.55/<0.05
H5N8-2016	-	-	<0.001	<0.01/<0.001	<0.001
H5N6-2017	-	-	-	<0.001	<0.001/0.13
H5N8-2020	-	-	-	-	<0.001
H5N1-2021	-	-	-	-	-

A single *p*-value cell means that the same *p*-value applies to both the D_T_ value and extinction time. - indicates that either (i) table diagonals: result not applicable as it is for the virus strain versus itself, or (ii) lower half of table: result available in the top-half of the table.

## Data Availability

All data are presented within this manuscript.

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
