# Peer review of "Assessment of Survival Kinetics for Emergent Highly Pathogenic Clade 2.3.4.4 H5Nx Avian Influenza Viruses"

_viruses, 2024, doi:10.3390/v16060889_

Round 1

Reviewer 1 Report

Comments and Suggestions for Authors

The authors investigated the thermal stability of H5 high pathogenicity avian influenza viruses (HPAIVs) in liquid (maybe cell medium). The difference between the past and current H5 HPAIVs on thermal stability is quite interest, but for in most of the authors, in molecular basis of HPAIVs nor epidemiology of HPAIV infection in wild birds and poultry. I agree that the contents of this manuscript should include the novelty on virology, however, the authors should refine the obtained data and add meaningful discussion more.

Overall, the authors developed the discussion without the results of statistical analyses. The authors insists the “difference on thermal stability”, however, unless indication on statistical analysis, the differences should not be confirmed. The authors must conduct the statistical analysis in all the data presented in the current work, and develop the discussion correctly based on the results of statistical analyses.

L60-65: This information is not directly related to the work of current study. It is so curious for me why the authors mention AIV stability under several environmental factors except temperature.

L72-74: We are not so sure about “SAPO” and “ACDP 3”.

L151-160: These descriptions are just explanations or background information, which may also be overlapped to Introduction already. It should be taken out.

L167-169: This sentence is so curious. The authors said that the incubation temperatures in this study were based on the setting where poultry may acquire infection. However, I have never heard that the poultry is fed at 4 0C.

L186-191: These discussions should be done based on the results of statistical analysis.

L193-203: This paragraph is meaningless for the current work and its outcomes. It is highly recommended to take it out.

L221-245: As well, this paragraph is meaningless for the current work and its outcomes. It is highly recommended to take it out.

L252-255: This discussion should be done based on the results of statistical analysis.

L276: There is less information about the black triangle. Which did temperature the authors put positive control?

L308: “with greater temperature stability” should be argued based on the results of statistical analysis.

L316-318: This is over-discussion. The current results just indicate the potential of thermal stability in liquid, not in other environment, like as soil, metal or wooden equipment.

Reviewer 2 Report

Comments and Suggestions for Authors

This study investigated the temperature stability of highly pathogenic avian influenza viruses isolated in the UK from 2014 to 2021. Although there are differences in temperature stability at 4 oC among virus strains, the variability of temperature stability at 4 oC diminishes as temperatures increase to 20 oC and 30 oC. The experimental data are properly conducted and convincing. However, the authors’ excessive discussions of the results and the experimental design that does not incorporate modern experimental techniques need to be reconsidered.

Major

1. L20-30 and subsequent discussion:

The authors propose that the emergence of temperature-stable virus strains may occur in warmer regions. This idea may be considered for the bacterial growth cycle, which does not require the infection of live animals. In general, viruses multiply in living cells, and the associated mutations also occur in the body of the infected animal. In other words, the emergence of viruses with different temperature sensitivity is the result of selection within the body of the infected bird. This phenomenon and the discussion that viruses with higher thermostability may be more resistant and finally predominant in the spread of infection are completely different messages. The text needs to be revised to avoid confusion.

2. If temperature stability continues to increase with each passing year when comparing viruses isolated from 2014 to 2021, the authors’ statements would be acceptable. However, the fact that the temperature stability of the virus in 2021 is lower than that of the other viruses was shown in this study. This may be because there are more important factors than temperature stability as a driving force for the global spread of the virus. For example, the influence of other factors such as the expansion of the virus' host range (such as the spread of infection not only in waterfowl living in freshwater, but also in seabirds) should be carefully explained.

3. This study is being conducted only with field strains isolated in the UK. Furthermore, no experimental results or discussion based on them regarding genetic differences among these viruses or functional differences in viral proteins have been described. Given current molecular biological techniques, the authors should generate genetic reassortant or mutants of the temperature-stable H5N8-2022 virus and the less temperature-stable H5N8-2014 virus to identify the viral proteins that contribute to this difference in temperature stability. Even if reverse genetics techniques are not established in your laboratory, it is possible to construct a similar experimental design using classical genetic reassortment techniques as follows.

a. Co-inoculate both viruses into the MDCK cells or chicken embryos.

b. Harvest propagated viruses and are incubated at higher temperatures for the selection of viruses with high thermostability.

c. Selection of higher thermostable clones by limited dilution or plaque cloning.

d. Sequencing and identification of key viral protein and its functional analysis.

Minor

4. Figures 1 and 2: The symbols for 4 oC and 30 oC are both the same and cannot be identified.

Round 2

Reviewer 2 Report

Comments and Suggestions for Authors

The revised manuscript has been properly revised in accordance with the reviewer's suggestions.